



# Determination of aerosol composition by ED-XRF on Teflon and quartz substrates: potentialities and limits

Florin Unga[1], Giulia Calzolai[2], Massimo Chiari[2], Eleonora Cuccia[3], Cristina Colombi[3], Mariolina Franciosa[3], Adelaide Dinoi[1], Eva Merico[1], Antonio Pennetta[1], Noelia Gómez-Sánchez[4], Caterina Mapelli[1], Salvatore Pareti[5], Cinzia Perrino[5], Eduardo Yubero[4], Daniele Contini[1]

[1]Institute of Atmospheric Sciences and Climate, ISAC-CNR, Lecce, 73100, Italy
[2]National Institute for Nuclear Physics – Florence Division, Sesto Fiorentino (FI), 50019, Italy
[3]ARPA Lombardia, Milano, 20124, Italy
[4]Atmospheric Pollution Laboratory, Miguel Hernández University. Elche, 03202, Spain
[5]Institute of Atmospheric Pollution Research, IIA-CNR, Monterotondo (RM), 00015, Italy

*Correspondence to*: Antonio Pennetta (antonio.pennetta@cnr.it)

**Abstract.** Energy Dispersive X-Ray Fluorescence (ED-XRF) is a versatile non-destructive technique to evaluate elemental composition of atmospheric particulate matter (PM) without the needs of sample preparation and with high potentiality in source apportionment studies. It is usually applied on Teflon or polycarbonate substrates; however, it would be preferable to use quartz substrates for the possibility to use the same substrate also for carbon detection. In this work an inter-comparison among five laboratories on $PM_{10}$ samples collected on Teflon and quartz filters was done with the specific purpose of understanding the performance of the ED-XRF technique applied to samples collected on quartz substrates. LODs on quartz substrates were significantly larger than those on Teflon for the majority of the elements with the exclusion of Cl, Mn, Cu, and Rb, which had comparable LODs for the two substrates. Repeatability on $PM_{10}$ samples collected on quartz and Teflon substrates was comparable and, on average, better than 10% for the majority of the elements analysed and better than 5% for several elements. Comparison of analysis on Teflon filters for twenty elements obtained by the different laboratories were in the range of ±15% of the 1:1 line for most of the elements and laboratories. Comparison of measurements on samples collected on quartz and Teflon substrates showed that 17 elements were well correlated (R>0.7) with average $C_{quartz}/C_{Teflon}$ ratios in the range 0.6±0.1 (for light elements, due to self-absorption effects) to 1.1±0.1 for the majority of the cases. This suggested that reasonable results could be obtained on quartz substrates for 17 elements, including Na, Mg, and Al, using calibration on Teflon and the ratios $C_{quartz}/C_{Teflon}$ as correction factors. However, these correction factors were dependent on the instrument and method used for the analysis.

**Keywords:** ED-XRF, PIXE, sampling substrates, aerosol composition, quartz fibre filters.



## 1 Introduction

Energy Dispersive X-Ray Fluorescence (ED-XRF) is a versatile technique to evaluate the elemental composition of atmospheric particulate matter (PM), which can be used with limited or negligible sample preparation (Calzolai et al., 2008; Canepari et al., 2009; Contini et al., 2016). Despite the limited sensitivity for trace elements, compared to other techniques such as ICP-MS, it has the advantages to measure crustal elements such as Si and Al providing useful information for source apportionment to characterise soil sources as well as resuspended dust in urban/suburban areas (Wang et al., 1999; Cesari et al., 2021). In addition, the technique is completely non-destructive and the collected samples (i.e. filters) can be successively used for other chemical analyses, being a significant advantage when different chemical approaches need to be used on the same samples or when chemical composition of soluble and insoluble PM have to be determined (Perrino et al., 2011).

Recent instrumental advances have enabled the possibility of high temporal resolution (~1 hour) multielement ED-XRF measurements which have proven to be useful for identification of specific natural and anthropogenic sources including soil dust (Furger et al., 2020), especially when coupled with other instruments such as Aerosol Chemical Speciation Monitor (ACSM), aethalometers and total carbon analysers (Manousakas et al., 2022). Measurements of elemental composition at high temporal resolution have been successfully applied in source apportionment analyses using receptor models (Belis et al., 2019; Su et al., 2020).

Offline ED-XRF is typically used for particulate matter (PM) samples collected on Teflon (Ogrizek et al., 2022) or polycarbonate substrates (Spolnik et al., 2005; Arana et al., 2014). However, in several monitoring stations and observational platforms, PM samples are routinely collected on quartz filters which allow the determination of organic and elemental carbon, also performing reasonably well for the determination of water-soluble ions via HPIC, metals via ICP-MS or other destructive techniques, and oxidative potential (Guascito et al., 2023). Thereby, it would be useful to use ED-XRF for elemental analysis on quartz substrates. ED-XRF on quartz suffers of absorption effects due to the penetration of the particles into the fibres of quartz, mainly affecting low energy X-rays, i.e. light elements (Chiari et al., 2018).

The studies focusing on ED-XRF performance and measurements on quartz substrates are yet relatively few. Chiari et al., (2018) investigated the absorption on quartz substrates by using ED-XRF and PIXE by comparing PM10 samples collected on Teflon and quartz substrates. The showed limits of detection (LODs) were significantly higher on quartz substrates, but good correlations between measurements on the two substrates, suggesting that average correction factors can be used for ED-XRF application on quartz substrates. However, the light elements (Na, Mg, Al, and Si) were not included in the analysis. Okuda et al. (2013) showed that the ED-XRF can be successfully applied to the determination of twelve elements (not including the light ones) on quartz filters. In Manousakas et al. (2013), ten elements were determined on PM10 samples collected on both substrates with significant greater uncertainty on quartz substrates compared to Teflon. Yatkin et al. (2012) compared standardless ED-XRF analysis of thirteen elements with ICP-MS analysis on Teflon and quartz filters and showed that, using appropriate correction factors, the results on quartz filters were also reasonable for Al. Similar conclusions were also obtained by Steinhoff et al. (2000) comparing ED-XRF analysis with atomic spectrometric methods (GF-AAS and ICP-OES). Gupta et al. (2021) compared measurements of seven elements using power-adapted wavelength dispersive WD-XRF with flame atomic absorption spectrometry (FAAS) on PM2.5 samples collected on quartz filters. Their results showed a good correlation for five elements (Al, Ca, Fe, K, and Na) and limited or negligible correlation for Ni and Zn.

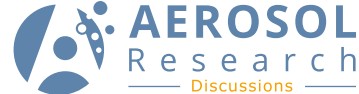

To address the gaps in current research, the main objectives of this work were: (i) to investigate limits of detection and repeatability of ED-XRF measurement of different elements contained in daily PM10 samples collected on both substrates (Teflon and quartz); (ii) to intercompare the concentrations of elements measured by ED-XRF and PIXE by different laboratories on daily PM10 samples collected on Teflon filters; (iii) to investigate the potentiality of ED-XRF to measure the concentration of 20 elements, including the light ones (i.e. Na, Mg, and Al), on quartz substrates, determining the correction factors necessary to adapt the calibration usually done for Teflon to quartz substrates.

## 2 Materials and methods

### 2.1 Details of the instruments used

$PM_{10}$ daily samples collected in various sites in Italy were analysed in the different laboratories involved in this work using their "home" instruments and protocols. Specifically: (i) ISAC-CNR (Lecce) used a Spectro (XEPOS05) ED-XRF instrument (Dinoi et al., 2024); (ii) ARPA Lombardia (Milan) used a Malvern Panalytical™ (Epsilon 4) ED-XRF spectrometer (Colombi et al., 2013); (iii) IIA-CNR (Rome) used a Spectro (XEPOS03) ED-XRF (Perrino et al., 2022a; 2022b); (iv) UMH (Elche) used a Thermo Scientific™ (ARL™ QUANT'X) ED-XRF spectrometer (Galindo et al., 2018); (v) INFN (Florence) used dedicated external-beam set-up for Particle Induced X-ray Emission (PIXE) analysis on aerosol samples, installed at the LABEC 3MV tandem accelerator (Calzolai et al., 2006).

Medium concentration elemental standards from Micromatter were used for calibration at ISAC-CNR, ARPA Lombardia, and UMH. ARPA Lombardia and UMH also used NIST2783 aerosol standard to routinely check instrumental performance. IIA-CNR calibrated on real samples by comparison with ICP-MS analysis (Astolfi et al., 2006; Canepari et al., 2009). INFN uses a standardless approach (Calzolai et al., 2006).

### 2.2 Sets of PM₁₀ samples used in the work

Several datasets of daily $PM_{10}$ collected at different sites on quartz and Teflon substrates (47 mm in diameter) using low-volume (i.e. 2.3 m³/h) samplers have been used in this work.

- A set of 30 daily $PM_{10}$ samples was collected by IIA-CNR in Rome in the research area of Montelibretti, a peri-urban site at 25 Km from Rome, in central Italy. Samplings were carried out in the period January 3 – February 5, 2023, on 47 mm Teflon filters, 2 µm pore size, (PALL Corporation, Port Washington, NY, USA) using a beta attenuation monitor operating at the flow rate of 2.3 m³/h (SWAM5a, FAI Instruments, Fonte Nuova, Rome, Italy).
- A set of 20 + 20 $PM_{10}$ daily samples were simultaneously collected by IIA-CNR on Teflon (same as above) and quartz (Pallflex Tissuquartz, PALL Corporation, Port Washington, NY, USA) substrates in an urban background site in Ferrara (Italy) using a dual channel beta attenuation monitor operating at the flow rate of 2.3 m³/h (SWAM5a Dual Channel Monitor, FAI Instruments, Fonte Nuova, Rome, Italy).
- A dataset of 9 $PM_{10}$ daily samples collected by Arpa Lombardia in Milan (north Italy) on Teflon filters and a set of 52 $PM_{10}$ daily samples simultaneously collected in an urban background site in Turbigo (Italy), about 35 km west of Milan city centre on Teflon (26 samples) and quartz (26 samples) substrates using a dual channel low-volume (i.e 2.3 m³/h) sampler (Gemini, Dado Lab srl).
- A dataset of 16 $PM_{10}$ samples collected simultaneously on quartz (8 samples) and Teflon (8 samples) substrates by ISAC-CNR in Lecce (south Italy) at the urban background site of the Environmental-Climate Observatory



(ECO) using a dual channel sampler at 2.3 m³/h (SWAM, Fai Instruments srl) equipped with automatic detection of concentrations by means of the β-attenuation method (Conte et al., 2020).

### 2.3 Determination of LODs and repeatability

The Limits Of Detection (LODs), for each element, on both Teflon and quartz filters, were evaluated as three times the standard deviation of element concentrations measured in at least six different field blanks for each substrate. For the elements not detectable on blanks, the LOD was evaluated as three times the minimum detectable level of the instruments.

Repeatability was determined by measuring of Micromatter standards (in the laboratories using them) and $PM_{10}$ samples collected on Teflon and quartz substrates. The repeatability on samples was obtained performing at least six different measurements (also in different days) by removing and reinstalling the samples in the analyser after each measurement.

### 2.4 Inter-comparison approach for Teflon and quartz substrates

The datasets described in Section 2.2 were analysed by the different laboratories using their typical "home protocol" and the concentrations of the different elements on daily $PM_{10}$ Teflon substrates were compared in ng/cm² after blank subtraction. Concentrations for As, Se, Mo, Cd, Te, and I were almost always lower than the LODs and these elements were excluded from the comparison. Ga and Ba were measurable in some samples but were included only in the ISAC-CNR protocol and were not further processed in this intercomparison. In total, twenty elements were considered in this comparison work: Na, Mg, Al, Si, P, S, Cl, K, Ca, Ti, Cr, Mn, Fe, Ni, Cu, Zn, Br, Rb, Sr, Pb.

Daily $PM_{10}$ samples collected simultaneously on Teflon and quartz substrates were analysed, with blank subtraction, by the different laboratory to estimate the correlations between measurements on the two substrates and the ratios of concentrations. This was done for all elements but Si. which was obviously not measurable on quartz filters. The aim was to determine correction coefficients to be used for determination of element concentration in quartz samples starting from the calibration of ED-XRF carried out on Teflon substrates. When concentrations on quartz substrates need to be determined, to normal protocol used for Teflon substrates can be applied and these average ratios used to correct measured concentrations after blank subtraction. This is possible for elements that are linearly correlated when detected on the two substrates, provided that the ratios can be determined with reasonably small standard errors. The latter because the final analytical uncertainty for measurements on quartz substrates will be the quadratic sum (assuming random independent errors) of the uncertainty resulting from the correction coefficients and that coming from the calibration on Teflon substrates.

### 3  Discussion of results

### 3.1 LODs for different substrates

The values of the LODs on both substrates averaged over the data from the different laboratories, are shown in Fig. 1 together with their standard errors. It was observed that the LODs of light elements were larger than those of heavy elements (i.e. high atomic numbers), especially for quartz filters. Ca is an exception to this trend showing an average LOD similar to that of Na for the quartz substrates and greater than that of Mg and Al for the Teflon substrates. For the majority of the elements the LODs for quartz filters were larger than those for Teflon filters with the exception of Cl, Mn, Cu, and Rb, which had LODs comparable values for the two substrates. A large inter-laboratory variability was observed for the LODs of some specific elements such as Na and Br (for both substrates); P, Ca, and Mn (for quartz substrates);



Zn (for Teflon substrates). This suggests, as expected, that the LODs depend on the instrument, on the specificity of the protocol used, and likely also on the filter batch.

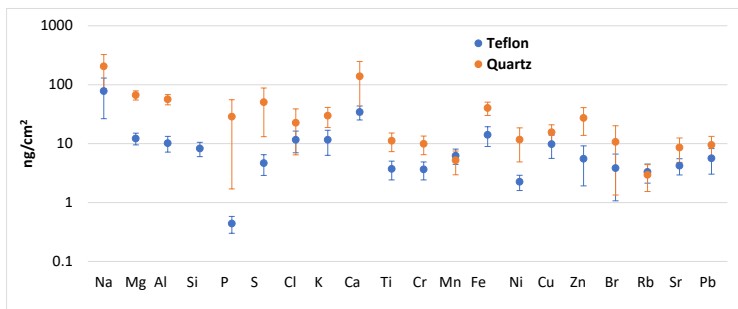

Fig. 1) Average LODs for ED-XRF analysis, of measurements on Teflon and quartz substrates by the different laboratories. The error bars are the standard errors of the results of the different laboratories.

### 3.2 Repeatability of measurements on standards and on PM$_{10}$ samples

The repeatability of measurements on medium concentration Micromatter standards was investigated at ISAC-CNR over

different time periods covering a range of 18 months to evaluate the long-term stability of calibration. Results are reported in Fig. 2 in terms of relative differences of measurements at 6, 9, 12, and 18 months from the initial calibration, taken as reference.

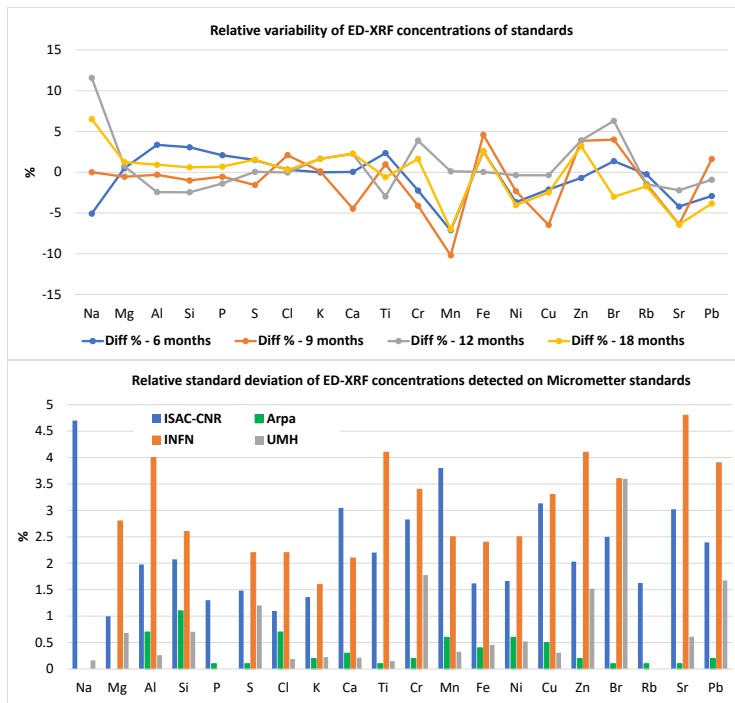

Fig. 2) (top) Repeatability (%) of measurements on standards at intervals of several months (6-9-12-18) from the first calibration at ISAC-CNR. (bottom) Average repeatability (%) of measurement on standards done by the different laboratories.

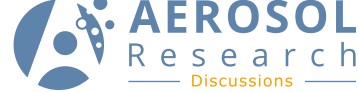

The data showed an excellent stability, with differences below 12% for all elements and below 5% (i.e. the nominal uncertainty of the standards) for most of the elements. In addition, there was not any clear evidence of a performance

deterioration with time because several differences at 18 months distance were lower (in absolute terms) than those at 3 months distance. This suggests that both the instrument and the standards are reasonably stable over long periods of time, such as those associated with the observational platforms of monitoring networks, and that periodic calibration checks done every few months may be sufficient for long term operation of ED-XRF. The average repeatability, between different tests, of elements measured on Micromatter standards by the different laboratories using them is also shown in Fig. 2,

with average values better than 5% for all elements.

Regarding repeatability on PM$_{10}$ samples, the results on the two substrates (i.e. Teflon and quartz) suggested comparable repeatability for all the elements measured. The results obtained by the different laboratories on Teflon substrates are reported in Fig. 3, showing repeatability better than 10% for most of the elements and laboratories and, in several cases, better than 5%. Sporadically (i.e. limited to specific laboratories) larger repeatability, in the range between

180 10% and 20%, was observed for Al, Cl, Ni, Cr, Rb, and Br.

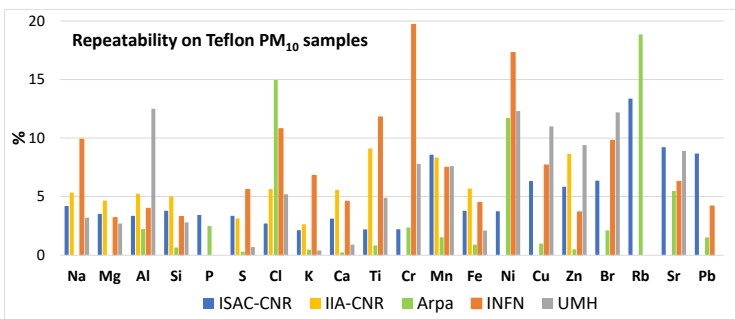

Fig. 3) Average repeatability (%) of ED-XRF measurements on PM$_{10}$ daily samples collected on Teflon filters by the different laboratories.

**3.3 Comparison of measurements on Teflon substrates**

The comparison was made on the Teflon samples of the datasets described in Section 2.2 comparing the results of the different laboratories on the same filters. The results of comparison for the non-trace elements (i.e. those with high abundances in the collected samples) are reported in Fig. 4, while the results for the trace elements are shown in Fig. 5.



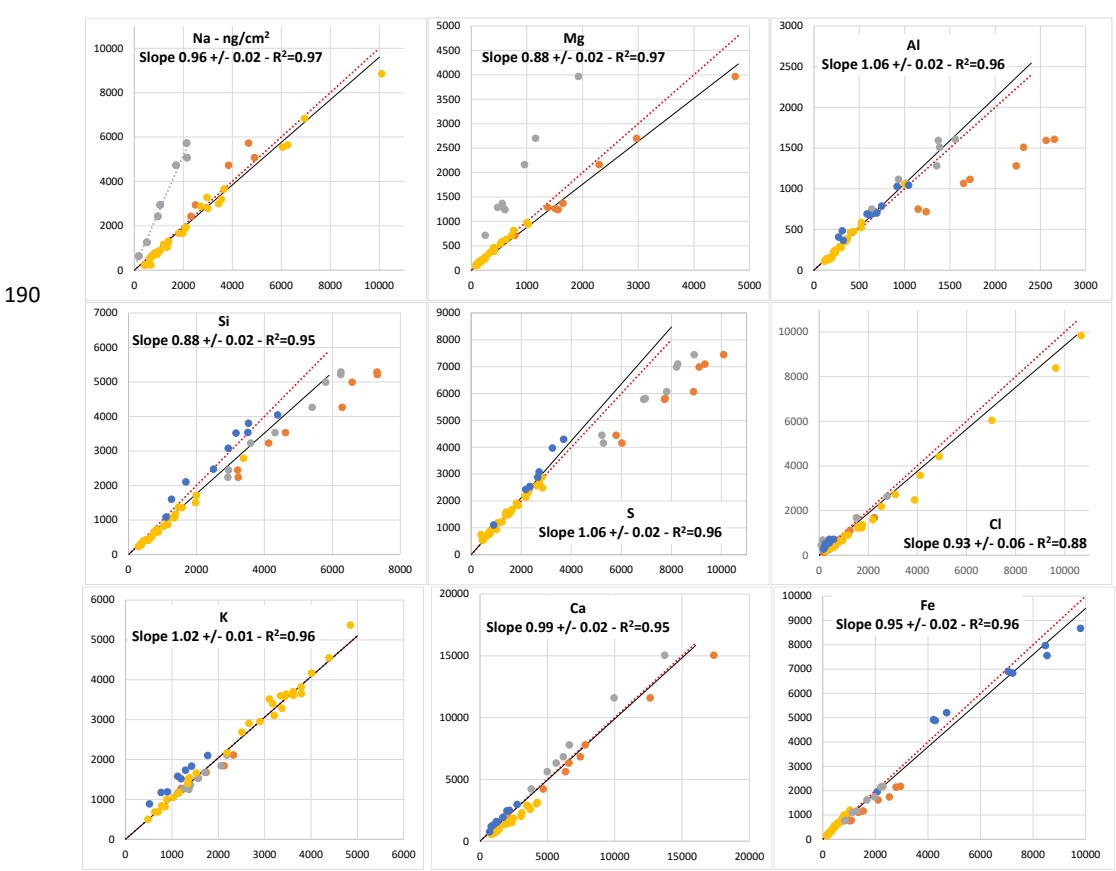

Fig. 4) Comparison of the measurements (ng/cm²) on Teflon filters for the non-trace elements. Red dashed line is 1:1, black continuous line is a linear fit including the datasets within 15% of the 1:1 line. X-axis reports measurements at IIA-CNR (yellow), Arpa (blue), UMH (grey); INFN with PIXE (orange); Y-axis reports measurements in Lecce for the corresponding samples.



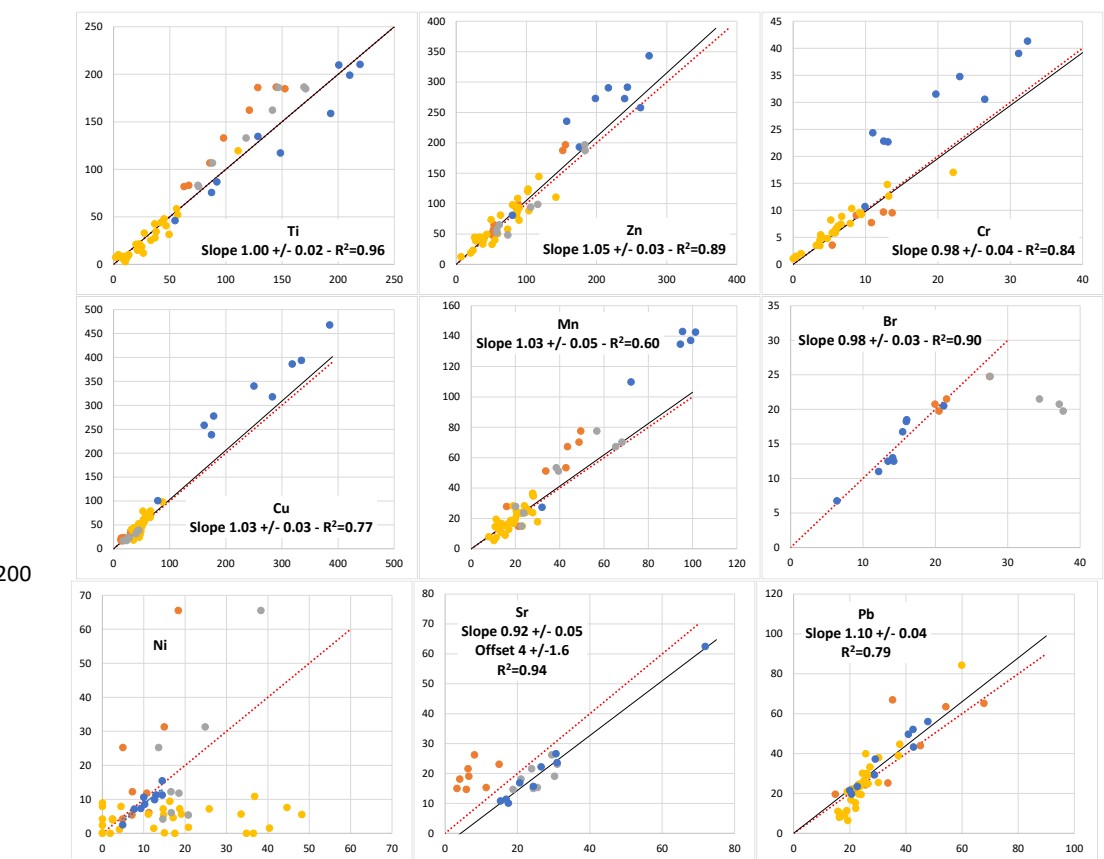

Fig. 5) Comparison of the measurements (ng/cm$^2$) on Teflon filters for the trace elements. Red dashed line is 1:1, black continuous line is a linear fit including the datasets within 15% of the 1:1 line. X-axis reports measurements at IIA-CNR (yellow), Arpa (blue), UMH (grey); INFN with PIXE (orange); Y-axis reports measurements in Lecce for the corresponding samples.

Regarding non-trace elements, all analysis for Na and Mg showed a linear behaviour with differences among the laboratories within 15% of the 1:1 line (with the exclusion of UMH for Na, showing an underestimation) that may suggest a difficulty to detect/calibrate these light elements with ED-XRF technique, if not specifically optimised. Concentrations of Al, Si, S with the ED-XRF analysis are typically within 15% of the 1:1 line, but an underestimation compared to PIXE was observed for Al (approximately 32%) and Si (approximately 28%). There were non-negligible differences between ED-XRF and PIXE also for S and Fe (both approximately 25% larger with PIXE), and Ti (approximately 20% lower with PIXE). Measurements of Cl showed a linear behaviour with inter-laboratory differences within 15% of the 1:1 line; the same happened for K with an offset of approximately 370 ng/cm$^2$ for the analysis of Arpa. Lombardia Measurements of Ca were within 15% of the 1:1 line for all laboratories with the exclusion of IIA-CNR, which showed an overestimation of approximately 25% compared to the average fitting line.

Regarding trace elements, Rb and P were only analysed by ISAC-CNR and Arpa Lombardia and are not included in Fig. 5. The Rb concentrations measured in the two laboratories were all less than 10 ng/cm$^2$ and correlated well (Pearson coefficient 0.94) with a slope of 0.92 ± 0.12, suggesting that even at these low concentration levels, Rb could be reliably measured. P concentrations of ISAC-CNR and Arpa Lombardia showed a very good correlation coefficient of 0.95 and a



slope of 0.76 ± 0.03, suggesting a potential non-negligible error in the determination/calibration of P with ED-XRF that should be verified and further investigated with other techniques. The other trace elements (Fig. 5) showed good linear behaviour when measurements of the different laboratories were compared, with the exclusion of Ni. For this element, there was a good correlation and slope of the measurements at ISAC-CNR and Arpa Lombardia (Pearson

coefficient=0.91; slope of 1.02 ± 0.17), but no correlation was found with those of the other laboratories. This result suggests that Ni may be an element that is difficult to be measured with ED-XRF technique at least at the low concentration levels of the samples of this study. Comparison of Zn and Pb measurements performed by the different laboratories showed linear behaviour with slopes reasonably near one. Br was detectable only in a limited number of samples and with low concentrations (i.e. < 30 ng/cm$^2$), however, measurements at the different laboratories had a linear

behaviour with slopes within 15% of the 1:1 ratio for ISAC-CNR, Arpa Lombardia, and INFN; however, no correlation was found with measurements at UMH. Cr showed a linear behaviour among measurements in different laboratories but an offset of approximately 9 ng/cm$^2$ was observed for Arpa Lombardia. The same was observed for Cu with an offset of approximately 50 ng/cm$^2$, at list at high concentrations (> 200 ng/cm$^2$). Sr measurements showed a good correlation between ISAC-CNR, UMH, and Arpa Lombardia, with an offset of approximately 4 ± 1.6 ng/cm$^2$ in the measurements

of ISAC-CNR. An underestimation of PIXE measurements compared to ED-XRF was observed for Sr. Measurements of Mn showed a substantial agreement of ISAC-CNR and IIA-CNR measurements; however, these showed an overestimation of approximately 26-28% compared to the other laboratories. This suggests that further comparison with independent chemical analysis of Mn using another technique such as the ICP-MS may be necessary.

The range of the average differences among measurements in the different laboratories and between ED-XRF

and PIXE were comparable with the results of some previous intercomparison studies. An inter-laboratory comparison of ED-XRF and PIXE was conducted by Gini et al. (2021) using certified reference materials deposited on Teflon filters. Their results showed an efficient detection of most of the elements (Ca, Fe, K, Ti, Zn, Cr, Pb), but only three participants were able to report values for light elements (i.e. atomic numbers <16). The average relative differences between the participants results and the assigned values were 17.5 ± 18.1% (reference material CRM2583; excluding Cr and Pb) and

16.7 ± 16.7% (reference material CRM2584; excluding Cr and P). Yatkin et al. (2016) performed an inter-laboratory comparison on PM$_{10}$ samples collected on Teflon filters at a regional rural site in north Italy, using three different XRF methods and the PIXE method. Regression results showed that the three XRF laboratories measured very similar mass loadings for S, K, Ti, Mn, Fe, Cu, Br, Sr and Pb, with slopes within 20% of unity.

**3.4 Comparison of simultaneous measurements on quartz and Teflon substrates**

The analysis of the PM$_{10}$ dataset collected simultaneously on quartz and Teflon filters was carried out to investigate the correlation between the concentrations measured on the two substrates and their ratio C$_{quartz}$/C$_{teflon}$ (where C$_{quartz}$ means concentration measured in the quartz filter and C$_{teflon}$ the concentration measured in the Teflon substrate). Results showed that Si was not measurable on quartz filters due to the excessive loading of the blanks; in addition, Ni, Rb, measured on

the two substrates showed no correlation, at the typical concentrations of the samples used in this study, and were not further processed. Results for Ni are consistent with the low correlation of Ni measurements on quartz substrates using WD-XRF with flame atomic absorption spectroscopy measurements observed in Gupta et al. (2021).

Figure 6 shows the average ratios C$_{quartz}$/C$_{Teflon}$ ratios for the different laboratories together with the inter-quartile range (i.e. 25$^{th}$ – 75$^{th}$ percentile).



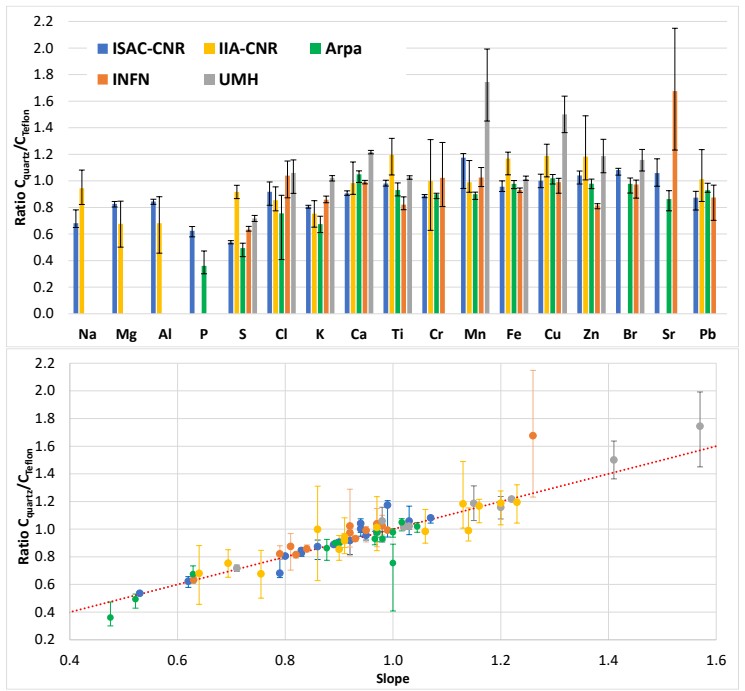

Fig. 6) (top) Averages and inter-quartile ranges (i.e. $25^{th}$ – $75^{th}$ percentiles) of the $C_{quartz}/C_{Teflon}$ ratios measured at the different laboratories. (bottom) Comparison of the averages $C_{quartz}/C_{Teflon}$ ratios with the corresponding slope of the linear fit between concentrations measured on quartz and Teflon. The red dashed line is 1:1. Error bars represent the inter-quartile ranges (i.e. $25^{th}$ – $75^{th}$ percentiles).

In addition, Figure 6 also compares the average ratios with the corresponding slopes of the linear fits between the concentrations measured on Teflon and on quartz substrates. Pearson correlation coefficients (not shown) were all above 0.7, and in several cases above 0.9, with the exception of Sr (0.64 at ISAC-CNR and 0.34 at INFN) with all concentrations below 30 ng/cm²; Br at INFN (0.34) with all concentrations below than 15 ng/cm²; Na (0.40), Cr (0.64), and Pb (0.49) at IIA-CNR. The comparison of the average ratios with the slopes showed a substantial agreement within the error bars for the different laboratories. For the PIXE analysis the only case out of the trend is the high value obtained on Sr but the two time-series were affected by significant uncertainty having all concentrations lower than 30 ng/cm² and a limited correlation (0.34) among measurements on quartz and Teflon. In a previous work, a ratio of 0.9 was observed for Sr using PIXE at different concentrations (Chiari et al., 2018). Regarding the average ratios observed at UMH there are two high values (> 1.4), obtained for Cu and Mn, which are significantly higher than the corresponding values of the other laboratories. The correction factors to be used for measurements on quartz substrates (i.e. the $C_{quartz}/C_{Teflon}$ ratios) were generally lower for light elements than for heavy elements. The trend observed is consistent with the results reported for twelve elements in Steinhoff et al. (2000), where a $C_{quartz}/C_{Teflon}$ ratio of about 0.6 was found for Al and K, and larger values of 1.0-1.1 were found for Zn, Pb, Cu, Fe, and Ca with ratios up to 1.2-1.4 for Ti and Se. The trend is also consistent with the results reported for eleven elements analysed by ED-XRF (excluding Al) in Chiari et al. (2018), where the values range from 0.6 for light elements (Na and Mg) to 0.7 (for S, Cl, and K) and up to 0.9-1.1 for heavier elements. The values of the ratios found in this work depend on the specific protocols and instruments used in the different laboratories, but for several of the 17 elements studied there is a good consistency with limited differences between the different laboratories.



In addition, reliable results could also be obtained on quartz substrates also for light elements (Na, Mg, and Al), provided that the instrument has sufficient emission energy and that the method used is optimised for these elements.

### 4 Conclusions

This work intercompared ED-XRF and PIXE measurements, done by different laboratories, on PM10 samples collected at different sites on quartz and Teflon substrates. The focus was to evaluate LODs, repeatability and correction factors to use ED-XRF on quartz substrates also for light elements (i.e. Na, Mg, Al, and P) starting from the calibration used for

Teflon substrates. The main findings are outlined below.

- LODs on quartz substrates were significantly larger than those on Teflon for the majority of the elements, with the exclusion of Cl, Mn, Cu, and Rb that had comparable LODs for the two substrates. LODs of light elements were generally larger compared to those of high atomic number, especially for quartz substrates, with the exception of Ca.

- Repeated measurements on Micromatter standards showed very stable results for several months (up to 18 months). The average repeatability of measurements of standards were better than 5% (i.e. nominal uncertainty of the standards) for all elements analysed. Repeatability on $PM_{10}$ samples collected on quartz and Teflon substrates was comparable and, on average, better than 10% for the majority of the elements analysed and better than 5% for several elements.

- Comparison of analysis on Teflon samples for twenty elements obtained by the different laboratories showed good correlation with concentrations in the range of ±15% of the 1:1 line for most of the elements and laboratories. Larger differences, 25%-30%, were observed comparing ED-XRF with PIXE for some elements (Al, Si, S, and Ti). Worse agreement between the different laboratories was observed for the determination of Ni, Mn, and Rb. Offsets > 10 ng/cm$^2$ were sporadically observed for Cu and K.

- Comparison of measurements on samples collected on quartz and Teflon substrates showed that on quartz Si was obviously too abundant on blanks and thus not measurable; Rb and Ni were not well correlated between the two substrates at the typical concentrations of this study. The remaining 17 elements were well correlated, with Pearson coefficients larger than 0.7 for the majority of the cases. The average ratios $C_{quartz}/C_{Teflon}$ of the majority of elements and laboratories were in the range 0.6±0.1 (for light elements, due to self-absorption effects) to

315 1.1±0.1 for heavy elements.

- Results showed that reasonable results could be obtained on quartz substrates for 17 elements, including Na, Mg, and Al for instruments having sufficient emission energy and a method optimised for these elements, using calibration on Teflon and opportune corrections based on the ratios $C_{quartz}/C_{Teflon}$. However, the corrections are depending on the instrument and method used for the analysis.

Further studies on the application of ED-XRF to multi-elemental measurements on quartz filter may benefits from the availability of suitable standards deposited on quartz substrates.

### Author contributions

DC conceptualized the study design and collaborated to the writing of the initial draft; FU, AD, EM, CM collected samples

in Lecce and performed ED-XRF analysis and some of the data post-processing; AP collaborated to measurements of elements and to the writing of the initial draft; GC, MC performed the analysis with PIXE; EC, CC, MF collected samples in Milano and performed ED-XRF analysis; SP, CP collected samples in Rome and performed ED-XRF analysis; NGS,



EY performed ED-XRF analysis in Elche. All authors collaborated to the interpretation of results, read and commented the final manuscript.

**Competing interests**

At least one of the (co-)authors is a member of the editorial board of Aerosol Research.

**Acknowledgements**

This research was supported by the project IR0000032 – ITINERIS, Italian Integrated Environmental Research Infrastructures System (D.D. n. 130/2022 - CUP B53C22002150006) Funded by EU - Next Generation EU PNRR - Mission 4 "Education and Research" - Component 2: "From research to business" - Investment 3.1: "Fund for the realisation of an integrated system of research and innovation infrastructures".

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
