# Peer review of "Determination of aerosol composition by ED-XRF on Teflon and quartz substrates: potentialities and limits"

_Aerosol Research, 2025_

## Author Response (AR2)

**RC1**

**General comment:**

**This paper uses EDXRF to study the chemical composition of particulate matter collected on quartz fiber and PTFE filters. While the results presented are not scientifically flawed, the content is too technical and contains almost no new scientific findings.**

Response:

We thank the reviewer for the constructive comment and for the time spent at reading our paper. We acknowledge that the manuscript has a technical focus, as it was conceived with the primary aim of evaluating the applicability of ED-XRF for the chemical characterization of PM collected on quartz fibre filters, which are commonly used in air quality monitoring for multi-parametric analysis, allowing the application of analytical protocols that cannot be performed on Teflon filters, such as the determination of organic and elemental carbon. While the work does not aim to propose new scientific results regarding aerosol sources or composition, we believe that the intercomparison exercise among different laboratories, the evaluation of detection limits, repeatability, and the assessment of correction factors for usage of ED-XRF on quartz substrates provide useful insights for the scientific community and for laboratories involved in routine PM chemical analysis.

**Specific comments:**

**L38 It is not appropriate to state in the introduction that X-ray fluorescence analysis is useful for Si measurements, despite the fact that the main purpose of this study is to use quartz fiber filters.**

Response: we thank the reviewer for the useful comment. We agree that the sentence may create ambiguity in the context of the study, where quartz fibre filters are used and Si cannot be measured due to the high blank levels of the substrate. However, the statement in the introduction was intended to describe the general potentialities of ED-XRF compared to other techniques such as ICP-MS, without yet addressing the specific choice of sampling substrate. Nevertheless, we acknowledge that this could lead to misunderstanding, and we will revise the sentence in the revised version of the manuscript as follows:

"*Despite the limited sensitivity for trace elements compared to other techniques such as ICP-MS, ED-XRF offers the advantage of being able to measure crustal elements such as Si and Al when suitable substrates are used, providing useful information for source apportionment to characterise soil sources as well as resuspended dust in urban and suburban areas.*"

**L41 The authors described that the EDXRF is completely non-destructive, but I believe that some of volatile compounds could be evapoled during EDXRF analysis.**

Response: we thank the reviewer for this useful observation. We agree that, although ED-XRF is generally considered a non-destructive technique, the possibility of losses of volatile species during the analysis cannot be completely excluded, especially for elements or compounds with high volatility under X-ray irradiation. However, in typical ED-XRF applications for ambient PM analysis, operating conditions (e.g., measurement at relatively low X-ray tube power) are designed to minimize this effect. In any case, we revised the sentence in the manuscript to clarify that *ED-XRF is a non-destructive technique with respect to the solid matrix of the sample, while potential losses of highly volatile species cannot be completely ruled out*.

**L54 What do you mean by "HPIC"?**

Response: We thank the reviewer for pointing this out. Indeed, we overlooked the explicit definition of the acronym "HPIC." We revised the sentence in the manuscript to clarify this point. The acronym will be expanded as follows: "HPIC (High-Performance Ion Chromatography)." The revised sentence will read: "*However, in several monitoring stations and observational platforms, PM samples are routinely collected on quartz filters which allow the determination of organic and elemental carbon, also performing reasonably well for the determination of water-soluble ions via High-Performance Ion Chromatography (HPIC), metals via ICP-MS or other destructive techniques, and oxidative potential.*"

**L95 I understand that this is depending on the researcher, but I feel that 2.3 m3/L is not a "low-volume", but a "middle volume" sampler.**
Response: we appreciate the reviewer's observation. While the classification of a sampler as "low-volume" or "medium-volume" can indeed be context-dependent, we have used the definition of "low-volume" in accordance with the European Standard EN 12341 (2014), which defines "low-volume" samplers as those with a flow rate of less than 5 $m^3$/h. Since the sampler used in this study operates at a flow rate of 2.3 $m^3$/h, we categorized it as "low-volume" based on this standard.

**Fig. 3, I'm wondering whether the repeatability depends on the mass loading on filter.**
Response: we thank the reviewer for the relevant observation. We agree that the repeatability of ED-XRF measurements can be affected by the mass loading on the filter, particularly for elements with concentrations close to the detection limits. In this study, the PM10 samples used for the repeatability evaluation were collected at different sites and at different environmental conditions thereby reflecting a certain heterogeneity in their composition and mass loading. The average repeatability shown was calculated using these samples with different mass loadings and we believe that it is representative of the typical repeatability obtainable in ambient air monitoring conditions.

**Fig. 4 In this scatter plot, the x-axis should indicate the reference or true value. I do not understand why only the Lecce sample is given special treatment. It has no scientific justification.**
Response: we thank the reviewer for the useful comment. If we had a reference or "true" values, it would have been used as x-axis. In this case, a reference value was not available and the choice of using measurements in Lecce as x-axis was done because it was the laboratory analysing all available samples, thereby representing a practical and reasonable choice for the x-axis but it was not intended to indicate a reference or true value. Besides, the figures allow a comparison among the different lab as well, for example, it was possible to put in evidence an underestimation of Lecce and Rome of Mn compared to the other labs that are in a better agreement one with the other.

**RC2**

**The authors presented the ED-XRF method for determining the elemental composition of atmospheric PM collected on quartz and Teflon filters. They included five laboratories in a comparative study of PM collected on quartz and Teflon filters and conducted an additional study comparing ED-XRF and PIXE measurements of Teflon filters.**

**The manuscript is well written but needs some clarifications and corrections.**

Response: We thank the reviewer for the constructive comments and for the time spent at reading our paper.

**Specific comments**

**Figures: Please correct all figures according to journal guidelines (e.g. (a), (b) notations, avoidance of titles, correct units such as ng cm⁻¹, etc.)**

Response: we thank the reviewer for this remark. All figures will be revised according to the journal guidelines.

**Please provide more details about the calibration. What concentrations or range of concentrations and how many points were used in (i) " medium concentration elemental standards from Micrometter" and (ii) real samples used for IIA-CNR calibration. ( Line 89)**

Response: we thank the reviewer for this suggestion. Medium concentration elemental thin-film standards from Micromatter (i.e. usually defined as light standards with mass loads in the range 6-50 $\mu g \ cm^{-2}$) were used for calibration at ISAC-CNR, ARPA Lombardia, and UMH using a single concentration point and linear calibration. ARPA Lombardia and UMH also used NIST2783 aerosol standard to routinely check instrumental performance. IIA-CNR calibrated on real samples by comparison with ICP-OES analysis (Astolfi et al., 2006; Canepari et al., 2009). Briefly, 20 Teflon filters loaded with atmospheric particles (about 55 $m^3$ sampled volume, collected in different environmental conditions) were analysed by ED-XRF and then re–analysed by ICP–OES, performing direct dissolution of the samples by HF acid digestion in microwave oven, applying the procedure described in Bettinelli et al. (2000). ICP-OES results on these ambient concentration samples were used for ED-XRF calibration, in addition to the original factory calibration values, which spanned over very high concentrations. In the course of time, the procedure was repeated by adding additional points in the calibration to reach 30 data points for each species (Astolfi et al., 2006; Canepari et al., 2009; Perrino et al., 2011). INFN uses a standardless approach (Calzolai et al., 2006), using the Micromatter thin elemental and the NIST2783 aerosol reference material as external standards for quality assurance and quality control checks.

**Please provide details of the instrument menthods and the »home« protocols used. (Lines 80-85)**

Response: we thank the reviewer for the useful comment. A summary of the procedures applied by the different laboratories are now included in the revised paper together with the bibliographic references where additional details can be found.

**Section 3.1: Only the average LODs of different laboratories are reported here. Is there a difference or a trend if you consider the laboratory as a variable and compare only the LODs of quartz and Teflon filters from the same laboratory?**

Response: thank you for this question. Figure 1 was built to give an overview of both, the LODs on the two substrates averaged for the different laboratories and the variabilities among the laboratories (i.e. the errors bars). It allows to discuss the trend observed with a focus on the two substrates. It is possible to compare each laboratory separately but this will results in five different figures ending with the same results with higher LODs on quartz substrates for the majority of the elements without furnishing additional information. For this reason, we prefer to maintain Fig.1 that bring this information in a compact way.

**Section 3.2: The first sentence states that repeatability was studied at ISAC-CNR, but Fig.2 (bottom) shows the average repeatability of many laboratories. Please clarify this section.**
Response: thank you for mentioning this aspect. To make things clearer, the original Figure 2 was divided into two figures in the revised paper. The long-term analysis on Micromatter standards was done only at ISAC-CNR (Fig. 2) while repeatability on samples was done by all laboratories (Fig. 3).

**Fig.2 ( bottom ): Can you clarify what is shown here? The title mentions relative standard deviation, but the caption says average repeatability.**
Response: the title of the figure has been eliminated; it is Fig. 3 in the revised paper (see the point above) and the caption was corrected. The figure reports the repeatability observed by the different laboratories on PM10 samples, calculated as reported in Section 2.3.

**Section 3.2: The results of the repeatability of PM10 samples on quartz filters are missing. Can you please include them.**
Response: thank you for spotting this. Repeatability tests on quartz filters was done only by three laboratories so that we preferred to eliminate this sentence in the revised paper.

**Line 187: Can you clarify what you mean by »the same filters«? Was the comparison done by different laboratories with the same filter batch or with the same PM samples?**
Response: thank you for spotting this unclear mentioning. It has been changed with "*the same PM$_{10}$ samples*".

**Line 235: What are »Sr. Measurements«?**
Response: we thank the reviewer for your observation. This will be rephrased to avoid confusion: "*An underestimation of PIXE measurements compared to ED-XRF was observed for strontium (Sr). Measurements of Mn showed a substantial agreement of ISAC-CNR and IIA-CNR measurements;*"

**Line 254: »...Si was not measurable on quartz filters due to the excessive loading of the blanks« Can you rephrase this statement? Si is usually not measured on quartz filters because quartz is made of Si and therefore causes interference, not because of excessive loading of the filter with PM.**
Response: we thank the reviewer for the constructive comment. The sentence will be modified as: "*Silicon was not measurable on quartz filters due to strong interference from the quartz matrix itself, which is composed primarily of Si and thus leads to high background levels.*"

**Lines 292-293 Have PIXE measurements also been performed on quartz filters? If so, can you include the results or explain why this was not the case?**

Response: yes, PIXE was done also on quartz filters because there was need to compare measurements on the two substrates. The results are reported in Figure 7 of the revised paper together with those obtained with ED-XRF by the other laboratories.

**The LOD of samples with quartz filters are quite high compared to other available methods for elemental determination. Given the high LODs, can you comment on whether this method is actually suitable for determining the elemental concentration of atmospheric PM? Is there any benefit or application where this method would be advantageous?**
Response: we thank the reviewer for the useful comment. This is a crucial point. While the LODs of ED-XRF are higher than those of other methods (such as ICP-MS), actually for both substrates Teflon and quartz, the technique still offers distinct advantages:

- Non-destructive analysis, allowing multi-method chemical characterisation on the same filter (e.g., carbon, ions, oxidative potential).
- Relatively fast and simple sample handling with minimal contamination risks.
- Sufficient sensitivity for many elements of interest for atmospheric PM levels.

We will add this discussion to the conclusions, highlighting scenarios where ED-XRF on quartz is especially beneficial, such as integrated observational platforms using a single filter type (e.g., quartz) for multiple co-located analyses.

"*Despite the higher LODs, ED-XRF on quartz filters remains advantageous in specific applications thanks to its non-destructive nature, fast and simple sample handling with minimal contamination risks, and its sufficient sensitivity for many elements relevant to atmospheric PM studies. These features make it particularly suitable for integrated platforms where multiple analyses (e.g., carbon, ions, oxidative potential) are performed on a single filter.*"